# Identification of the Ovine Keratin-Associated Protein 21-1 Gene and Its Association with Variation in Wool Traits

**DOI:** 10.3390/ani9070450

**Published:** 2019-07-16

**Authors:** Shaobin Li, Huitong Zhou, Hua Gong, Fangfang Zhao, Jiqing Wang, Xiu Liu, Jiang Hu, Yuzhu Luo, Jon G.H. Hickford

**Affiliations:** 1Gansu Key Laboratory of Herbivorous Animal Biotechnology, Faculty of Animal Science and Technology, Gansu Agricultural University, Lanzhou 730070, China; 2International Wool Research Institute, Gansu Agricultural University, Lanzhou 730070, China; 3Gene-Marker Laboratory, Faculty of Agricultural and Life Sciences, Lincoln University, Lincoln 7647, New Zealand

**Keywords:** KAP21-1 gene (*KRTAP21-1*), variation, wool yield, sheep

## Abstract

**Simple Summary:**

Keratin-associated proteins (KAPs) are key constituents of wool and hair fibers. In this study, the ovine keratin-associated protein 21-1 gene (*KRTAP21-1*) was identified, and association analysis showed that its variation affected wool yield, which prompted the conclusion that the gene may have potential use as a genetic maker for improving wool yield.

**Abstract:**

Keratin-associated proteins (KAPs) are key constituents of wool and hair fibers. In this study, an ovine KAP gene encoding a HGT-KAP protein was identified. The gene was different from all of the HGT-KAP genes identified in sheep, but was closely related to the human KAP21-1 gene, suggesting that it represented the unidentified ovine *KRTAP21-1*. Four variants (named *A* to *D*) of ovine *KRTAP21-1* were found in 360 Merino × Southdown-cross lambs from four sire lines. Three sequence variations were detected among these variants. Two of the sequence variations were located upstream of the coding region and the remaining one was a synonymous variation in the coding sequence. Six genotypes were found in the Merino-cross lambs, with only two of the genotypes (*AA* and *AC*) occurring at a frequency of over 5%. Wool from sheep of genotype *AA* had a higher yield than that from *AC* sheep (*p* = 0.014), but tended to have a lower greasy fleece weight (GFW) than that of genotype *AC* (*P* = 0.078). This suggests that variation in *KRTAP21-1* affects wool yield and the gene may have potential for use as a genetic maker for improving wool yield.

## 1. Introduction

Wool keratin-associated proteins (KAPs) are structural components of the wool and hair fibers. KAPs form a matrix that embeds the keratin intermediate filaments (KIFs) and they are believed to play a key role in defining the physico-mechanical properties of wool fibers [1]. The KAPs either possess a high content of cysteine, or are rich in glycine and tyrosine; and based on this, they have been categorized into three groups: the high glycine-tyrosine (HGT) KAPs, the high sulphur (HS) KAPs, and the ultra-high sulphur (UHS) KAPs [2].

HGT-KAPs are principally found in the orthocortex of the wool fiber and they are the first KAPs expressed after the synthesis of KIFs. The HGT-KAP content varies considerably in different wools, ranging from less than 1% by weight in the wool of Lincoln sheep, to up to 12% by weight in the wool of Merino sheep [3]. The wide range in the relative content of HGT-KAPs in wool from different breeds, and the low content of HGT-KAPs in sheep with the felting luster mutation [4], raises intriguing questions about the function of these proteins in determining wool fiber characteristics.

There are seven HGT-KAP families (KAP6–KAP8 and KAP19–KAP22) that have been identified in humans [5], but only six KAP families have been described in sheep, KAP6–KAP8, KAP19, KAP20 and KAP22 [2,6,7,8]. For some of the families (KAP6 and KAP8), there appear to be more members found in sheep than in humans [9,10], suggesting sheep may have more HGT-KAP genes, despite there being no report to date identifying the KAP21 family.

A recent bioinformatics analysis of the sheep genome assembly in proximity to *KRTAP8-2* [9], led to the discovery of a potential new open reading frame (ORF) (NC_019458.2: 122927759 to 122928094) that would encode a HGT-KAP (unpublished data). In this study, we describe the identification of this ORF, report sequence variation in it in different sheep as detected by polymerase chain reaction-single stranded conformational polymorphism (PCR-SSCP) analysis, and reveal associations between variation in the gene and variation in some wool traits.

## 2. Materials and Methods

All research involving animals were carried out in accordance with the Animal Welfare Act 1999 (New Zealand Government) and the collection of sheep blood drops by nicking sheep ears were specifically addressed by Section 7.5 Animal Identification, of the Animal Welfare (Sheep and Beef Cattle) Code of Welfare 2010; a code of welfare issued under the Animal Welfare Act 1999.

### 2.1. Sheep Blood and Wool Samples

Three hundred and sixty Merino × Southdown-cross lambs farmed at Ashley Dene (Lincoln University, Canterbury, New Zealand), and from four sire-lines, were used to search for sequence variation and to test associations with various wool traits.

All 360 lambs were ear-tagged with unique identification number within 12 h of birth and their birth dates, birth weights, birth ranks (i.e., whether they were a single, twin or triplet), genders and sire identity were recorded. All of the ewes and lambs were brought together at tail docking (lambs aged between 2–6 weeks old) and remained together until weaning. At tailing, blood samples from all these sheep were collected onto FTA cards (Whatman BioScience, Middlesex, UK) and genomic DNA was purified using a two-step procedure described by Zhou et al. [11].

Greasy fleece weight (GFW) was measured at shearing and other wool traits were measured by the New Zealand (NZ) Wool Testing Authority Ltd. (Ahuriri, Napier, New Zealand). The traits measured included mean fiber diameter (MFD), fiber diameter standard deviation (FDSD), coefficient of variation of fiber diameter (CVFD), prickle factor (PF, percentage of fibers over 30 microns), mean staple length (MSL), mean fiber curvature (MFC), mean staple strength (MSS) and wool yield (Yield). Clean fleece weight (CFW) was calculated from GFW and Yield (i.e., CFW = GFW × Yield).

### 2.2. PCR Primers and Amplification

The genome sequences flanking the newly identified ORF (NC_019458.2: 122927759 to 122928094) were used to design PCR primers that would amplify a fragment containing the ORF and that had a predicted size of 454 bp. The sequences of these primers were: 5′-AGAAGACACACACACTTCAG-3′ and 5′-CATTATCTTTAGAGGCTTCATC-3′. The primers were synthesized by Integrated DNA Technologies (Coralville, IA, USA).

PCR amplification was performed in a 15-μL reaction containing the genomic DNA on one 1.2-mm punch of FTA paper, 2.5 mM of Mg^2+^, 0.25 μM of each primer, 150 μM of each dNTP (Bioline, London, UK), 0.5 U of *Taq* DNA polymerase (Qiagen, Hilden, Germany), and 1× the reaction buffer supplied with the enzyme. The thermal profile consisted of an initial denaturation step of 2 min at 94 °C, followed by 35 cycles of 30 s at 94 °C, 30 s at 60 °C and 30 s at 72 °C, and with a final extension of 5 min at 72 °C. Amplification was carried out using S1000 thermal cyclers (Bio-Rad, Hercules, CA, USA).

### 2.3. Screening for Sequence Variation

The PCR amplicons were screened for sequence variation using SSCP analysis. A 0.7-μL aliquot of each amplicon was mixed with 7 μL of loading dye (98% formamide, 0.025% xylene-cyanol, 10 mM EDTA, 0.025% bromophenol blue). After denaturation at 95 °C for 5 min, the samples were rapidly cooled on wet ice and then loaded on 16 cm × 18 cm, 14% acrylamide: bisacrylamide (37.5:1) (Bio-Rad) gels. Electrophoresis was performed using Protean II xi cells (Bio-Rad) in 0.5× TBE buffer, under the electrophoretic conditions of 19 °C, 280 V for 16 h. Gels were silver-stained according to the method of Byun et al. [12].

### 2.4. Sequencing of the Variants and Sequence Analysis

PCR amplicons representing different SSCP banding patterns from sheep that appeared to be homozygous were sequenced using Sanger sequencing in both directions at the Lincoln University DNA sequencing facility, New Zealand. Variants that were only found in heterozygous sheep were sequenced using an approach described by Gong et al. [13]. Briefly, a band corresponding to the variant was cut as a gel slice from the polyacrylamide gel, macerated, and then used as a template for re-amplification with the original primers. This second amplicon was then sequenced directly as described above for the homozygous patterns.

Nucleotide sequence alignments and translation to amino acid sequences were undertaken using DNAMAN (version 5.2.10, Lynnon BioSoft, Vaudreuil, QC, Canada).

### 2.5. Statistical Analyses

Statistical analyses were performed using Minitab version 16 (Minitab Inc., State College, PA, USA). General Linear Mixed Models (GLMMs) were used to compare the various wool traits in sheep of different genotypes, and with a Bonferroni correction being applied to reduce the chances of obtaining false positive results during the repeated comparisons. Sire was found to affect (*p* < 0.05) all of the wool traits, and gender was also found to affect (*p* < 0.05) or potentially affect (*p* < 0.20) the wool traits; hence both sire and gender were included as explanatory factors in the models. Birth rank was not found to affect (*p* > 0.05) or potentially affect (*p* > 0.20) wool traits, and thus was not factored into the models.

## 3. Results

### 3.1. Identification of Ovine KRTAP21-1

The PCR primers amplified a fragment of DNA from sheep genomic DNA, but the length of the amplicon was only 391 bp which was 63 bp shorter than expected. Analysis of the amplicons with SSCP, revealed four distinct banding patterns that were produced by four different DNA sequences. These were named variants *A* to *D* (Figure 1A). There were three nucleotide sequence differences among these sequences, one being 11 bp upstream of the start codon, one being 1 bp upstream of the start codon, and the remaining one being a synonymous nucleotide substitution c.114C/T (Figure 1B).

All of these four sequences had an apparent loss of 63 bp in the middle of the coding region, when compared to the Texel sheep genome assembly sequence v.4 NC_019458.2, but this sequence is missing in the Rambouillet construct v.1 GCA_002742125.1. There were also other differences between the sequences identified in this study and the Texel sheep genome assembly sequence.

The ovine sequences were different from all of the sheep HGT-KAP genes identified to date, but were phylogenetically related to human KAP21-1 (Figure 2). This suggests that these ovine sequences represent ovine KAP21-1. This gene was physically located between *KRTAP8-2* and *KRTAP20-2*, and clustered with twelve other previously identified KAP genes on chromosome 1.

### 3.2. Association of KRTAP21-1 Genotypes and Wool Traits

Six genotypes were detected in the Merino × Southdown-cross sheep, and these were *AA* (*n* = 216, 60%), *AB* (*n* = 12, 3.3%), *AC* (*n* = 116, 32.2%), *AD* (*n* = 6, 1.7%), *BC* (*n* = 3, 0.8%) and *CC* (*n* = 7, 1.9%). The variant frequencies were 78.6%, 2.1%, 18.5% and 0.8% for variants *A*, *B*, *C* and *D*, respectively.

Only two genotypes (*AA* and *AC*) occurred at a frequency over 5% in the sheep investigated. With these two common genotypes, an effect of genotype was observed for yield and GFW (Table 1). Wool from *AA* sheep had a higher yield than that of *AC* sheep (*p* = 0.014), but tended to have a lower GFW than that of genotype *AC* (*p* = 0.078). No other associations were detected.

## 4. Discussion

This study has identified a new KAP gene that is located on sheep chromosome 1, and describes sequence variation in the gene and its association with Yield and GFW in Merino-cross sheep. The gene is different from all of the known KAP genes identified to date in sheep, clustered with other KAP genes, polymorphic, and contains a single exon that encodes a glycine and tyrosine-rich protein, suggesting it represents a previously unidentified ovine HGT-KAP gene. At the sequence level, it is most closely related to human *KRTAP21-1,* and it is located in an ovine chromosome region that is similar to where the human KAP21 genes are found. This allows us to be more confident that this is ovine *KRTAP21-1*. The identification of the ovine *KRTAP21-1* increases the number of HGT-KAP genes identified in sheep from 11 to 12.

The ovine *KRTAP21-1* sequences were very similar to an expressed sequence tag (EST) (GO704545) from a sheep skin cDNA library, with variant *D* having only two nucleotide differences to the EST in the 5′ region. These nucleotide differences may reflect unidentified sequence variation, or may be the result of amplification or sequencing errors. This suggests that the *KRTAP21-1* is expressed in sheep skin.

As found with the *KRTAP21-1* sequences reported here, the EST sequence (GO704545) also has a 63-bp sequence ‘missing’ in the middle of the gene, when compared to the Texel sheep genome assembly v.4 (NC_019458.2), although this sequence was missing in the Rambouillet construct v.1 (GCA_002742125.1). Given that only a small number of sheep from several breeds in NZ were investigated here, it is possible that this 63-bp sequence may reflect a breed or allelic difference, as non-frame shift insertions/deletions have been described in *KRTAPs*, such as *KRTAP5-4*, *KRTAP6-1, KRTAP6-3*, *KRTAP6-5* and *KRTAP20-1* [10,14,15]. The possibility also exists that this 63-bp sequence may have come about because of sequence assembly errors.

The finding that wool from *AC* sheep had a lower Yield and tended to have higher GFW than *AA* sheep, but with no difference in CFW, suggests that the gene does not affect the quantity of fiber proteins, but instead either affects the quantity of non-fiber wool components, such as the amount of wool wax, grease, vegetable matter or dirt, or the way in which those non-fiber components are retained in the fleece. While it is hard to comprehend how a fiber protein gene could affect the content of a non-fiber wool component, it is possible that the gene affects some intrinsic fiber property and consequently affects the yield without affecting the fiber weight. Two possible wool fiber properties affected might include the ability of the fiber to absorb water, and the shape of the fiber or specifically the surface area to volume ratio.

The KAP proteins are rich in hygroscopic amino acids (glycine and serine) which endow wool fibers with the ability to absorb over 30% of their weight in water before they begin to feel wet. The cortex of fine wool fibers has a bilateral structure and the orthocortical portion of the fiber is found to absorb water-based dyes more intensely than the paracortical portion [16], suggesting that the ratio of orthocortex and paracortex in individual fibers affects the ability to absorb water. The novel KAP21-1 protein would belong to the HGT-KAP group and this group of proteins is thought to be predominantly present in the orthocortex of the wool fiber [17]. Despite the variation identified in ovine *KRTAP21-1* not leading to changes in the amino acid sequence, the sequence variation in proximity to the start codon may affect gene expression. The synonymous coding sequence SNP may also have a functional effect, as it has been reported that synonymous SNPs can affect re-translation rates and co-translational protein folding [18].

Variation in the shape of the wool fibers might also affect the yield. Some wool fibers may be close to cylindrical in cross-section, while others may be more oval-like. Fibers that are more oval in cross-section will have an increased surface area to volume ratio. This increased surface area, relative to volume, may therefore provide more surface area for wool wax, suint and vegetable matter to accumulate on, thereby affecting GFW and yield. Unfortunately, cross-sectional fiber shape cannot be assessed easily or cost-effectively for a large numbers of fibers, as the best method for assessing the trait is electron microscopy.

The associations detected for *KRTAP21-1* are similar to those observed for *KRTAP15-1* [19] and *KRTAP22-1* [8], but different from those reported for *KRTAP6-1* [10], *KRTAP6-3* [20], *KRTAP20-1* [15], *KRTAP20-2* [7] and *KRTAP26-1* [21]. Physically *KRTAP21-1* is closer to *KRTAP20-2*, then *KRTAP6-3*, *KRTAP22-1* and *KRTAP6-3*, followed by *KRTAP20-1* and *KRTAP15-1*, and finally *KRTAP26-1* (Figure 3). The association reported for *KRTAP21-1* does correlate with the physical distance along the chromosome to the other *KRTAPs*, suggesting the association is unlikely to be due to the linkage with any other nearby *KRTAP*, and probably represents the effect of *KRTAP21-1* itself. Regardless, the detection of an association between *KRTAP21-1* and yield suggests that *KRTAP21-1* may have potential as a gene-marker for improving wool production.

## 5. Conclusions

Four variants of ovine *KRTAP21-1* were identified in Merino × Southdown-cross lambs. Three sequence variations were detected among these variants. Association analysis revealed that the variation in *KRTAP21-1* was associated with variation in yield, which suggests the gene may have potential for use as a genetic marker for improving this trait.

## Figures and Tables

**Figure 1 animals-09-00450-f001:**
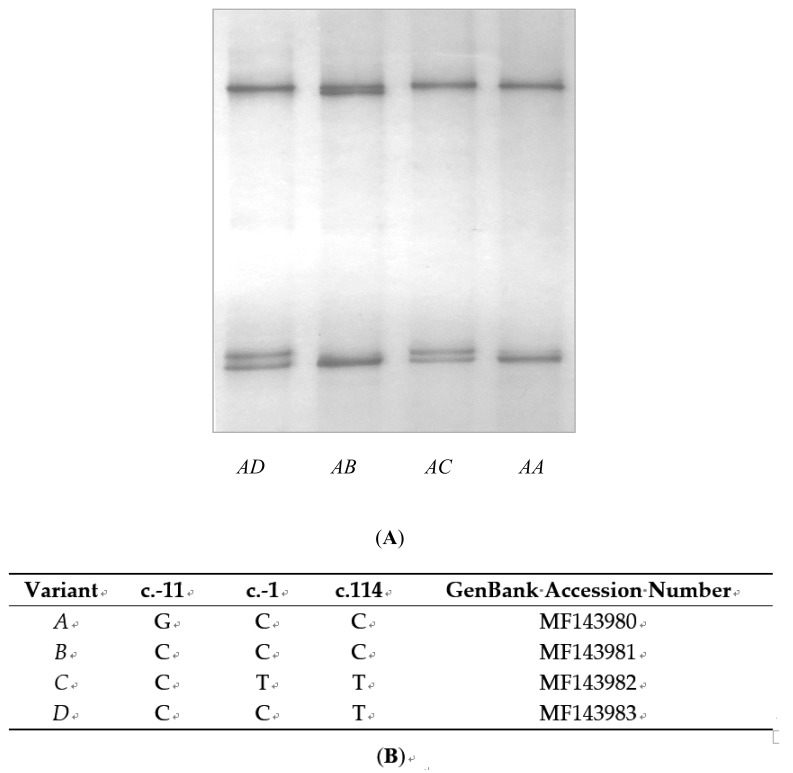
Variation in ovine *KRTAP21-1*. (**A**) Four unique polymerase chain reaction-single stranded conformational polymorphism (PCR-SSCP) banding patterns representing four variants (*A* to *D*) were detected. (**B**) Three substitutions were observed in the nucleotide sequences of the variants.

**Figure 2 animals-09-00450-f002:**
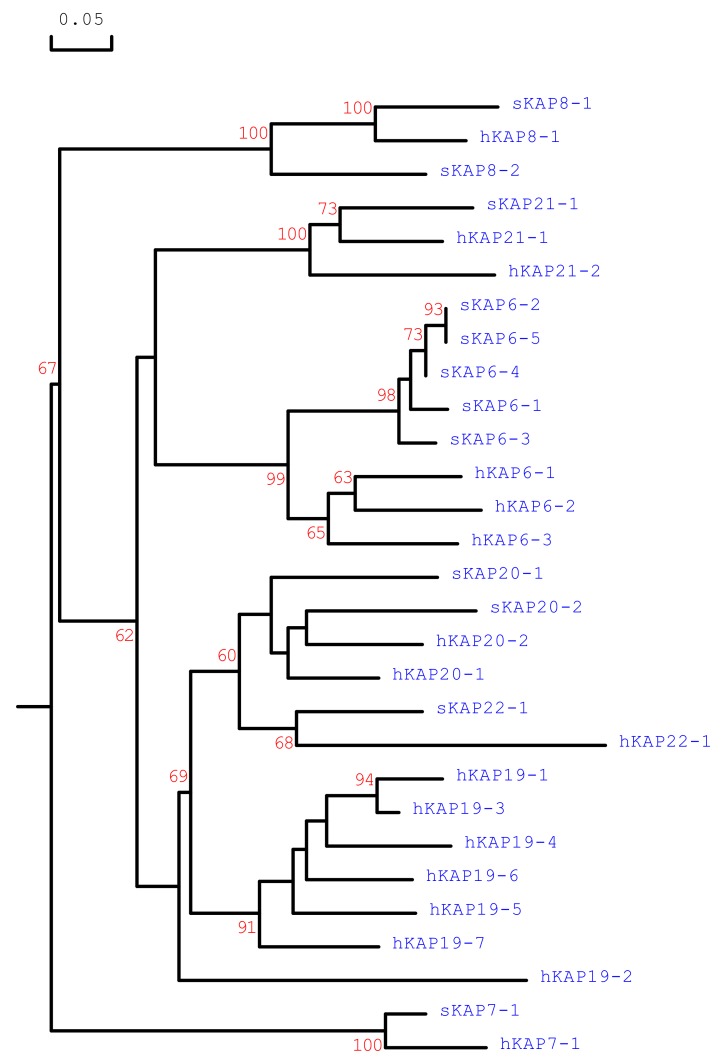
Phylogenetic tree of the newly identified ovine KAP21-1 along with other HGT-KAPs from sheep and humans. The tree was constructed using the predicted amino acid sequences. The numbers at the forks indicate the bootstrap confidence values and only those equal to, or higher than 60% are shown. Only one of the sheep KAP21-1 variants is shown as all of the four variants are identical at the amino acid sequence level. The sheep KAPs are indicated with a prefix “s”, while the human sequences are indicated with “h”. The GenBank accession numbers for the ovine HGT-KAPs are: NM_001193399 (sKAP6-1), KT725832 (sKAP6-2), KT725837 (sKAP6-3), KT725840 (sKAP6-4), KT725845 (sKAP6-5), X05638 (sKAP7-1), X05639 (sKAP8-1), KF220646 (sKAP8-2), MH243552 (sKAP20-1), MH071391 (sKAP20-2), and KX377616 (sKAP22-1). The GenBank accession numbers for the human HGT-KAPs are: NM_181602 (hKAP6-1), NM_181604 (hKAP6-2), NM_181605 (hKAP6-3), AJ457063 (hKAP7-1), AJ457064 (hKAP8-1), AJ457067 (hKAP19.1), NM_181608 (hKAP19-2), NM_181609 (hKAP19-3), NM_181610 (hKAP19-4), NM_181611 (hKAP19-5), NM_181612 (hKAP19-6), NM_181614 (hKAP19-7), NM_181615 (hKAP20-1), NM_181616 (hKAP20-2), NM_181619 (hKAP21-1), NM_181617 (hKAP21-2), and NM_181620 (hKAP22-1).

**Figure 3 animals-09-00450-f003:**
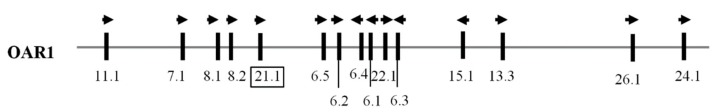
Location of *KRTAP21-1* and other *KRTAPs* on ovine chromosome 1. The newly identified ovine *KRTAP21-1* is shown in a box, along with the fourteen previously identified *KRTAPs*. The vertical bars represent the location of the different *KRTAPs* and the arrowheads indicate the direction of their transcription. The numbers below the bars indicate the name of the respective KAP genes (i.e., 24.1 is *KRTAP24-1*).

**Table 1 animals-09-00450-t001:** Effect of *KRTAP21-1* genotype on various wool traits.

Trait ^1^	Mean ± SE ^2^		*p*
	*AA* (*n* = 169)	*AC* (*n* = 99)	
GFW (kg)	*2.33 ± 0.04*	*2.42 ± 0.04*	*0.078*
CFW (kg)	1.71 ± 0.03	1.73 ± 0.04	0.656
Yield (%)	**72.6 ± 0.49**	**70.9 ± 0.59**	**0.014**
MSL (mm)	84.3 ± 1.72	85.4 ± 1.94	0.462
MSS (N/ktex)	21.6 ± 1.13	21.1 ± 1.28	0.557
MFD (µm)	19.3 ± 0.27	19.6 ± 0.31	0.361
FDSD (µm)	4.14 ± 0.09	4.15 ± 0.10	0.882
CVFD (%)	21.3 ± 0.34	21.1 ± 0.38	0.768
MFC (^o^/mm)	88.6 ± 2.28	89.2 ± 2.57	0.768
PF (%)	2.50 ± 0.50	2.86 ± 0.56	0.405

^1^ GFW—Greasy Fleece Weight; CFW—Clean Fleece Weight; MSL—Mean Staple Length; MSS—Mean Staple Strength; MFD—Mean Fiber Diameter; FDSD—Fiber Diameter Standard Deviation; CVFD—Coefficient of Variation of Fiber Diameter; MFC—Mean Fiber Curvature; PF—Prickle Factor (percentage of fibers over 30 microns). ^2^ Predicted means and the standard errors of those means derived from the GLMMs, with *p* < 0.05 in bold and 0.05 ≤ *p* < 0.10 italicized.

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
