# Peer review of "Identification of the Ovine Keratin-Associated Protein 21-1 Gene and Its Association with Variation in Wool Traits"

_animals, 2019, doi:10.3390/ani9070450_

Round 1

Reviewer 1 Report

The paper is largely acceptable in its current form after the authors have taken into consideration the following points:

Line 52 – The KAP19 family has been reported in a proteomic study conducted on Australian Merino sheep (see Almeida et al., J. Proteomics 103, 170-177, 2014)

Lines 211-212 – This effect may not be due to the better ability of orthocortical cells to absorb water than paracortical cells. The paracortical cells are often less accessible to stains because there is less intermacrofibrillar material than the orthocortical cells.  They retain stains better than orthocortical cells for the same reason. This difference disappears following enzymic removal of the intermacrofibrillar material.

Author Response

POINT 1: Line 52 – The KAP19 family has been reported in a proteomic study conducted on Australian Merino sheep (see Almeida et al., J. Proteomics 103, 170-177, 2014)

Response 1: We have amended the text.

POINT 2: Lines 211-212 – This effect may not be due to the better ability of orthocortical cells to absorb water than paracortical cells. The paracortical cells are often less accessible to stains because there is less intermacrofibrillar material than the orthocortical cells. They retain stains better than orthocortical cells for the same reason. This difference disappears following enzymic removal of the intermacrofibrillar material.

Response 2: This is a useful addition to the discussion, and while we cannot find a reference XXXX reference???XXX

Reviewer 2 Report

The manuscript describes a previously unidentified ovine KAP gene (KRTAP21-1) encoding a HGT-KAP protein. Four variants were identified in the gene in 360 Merion x Southdown lambs, giving rise to six different genotypes in total. These genotypes were then associated with different wool quality traits including fibre diameter, fleece weight and yield. The authors conclude that KRTAP21-1 has potential for use as a gene-maker for improving wool yield.

The research presented is interesting and well described. I have only a few minor comments to be addressed.

Line 79 change ‘samples’ to ‘sample’

Line 85 what was the final concentration of DNA added to each reaction?

Line 102 Add ‘by Sanger sequencing’ after ‘sequenced’ assuming this was the sequencing method used.

Line 136 The analysis here uses the Texel Oar_v4 assembly, there is a newer more contiguous assembly for sheep Rambouillet v1 GCA_002742125.1, have the authors compared with this assembly? Presumably the difference if not related to the genome assembly could be a breed specific difference?

Line 194 Could the authors clarify here in text that they are using the Oarv4 assembly. They could also make the point that the 63bp sequence might be resolved in the new assembly but this needs to be checked.

Line 231 Reference to figure 3 could be removed.

Author Response

POINT 1: Line 79 change ‘samples’ to ‘sample’

Response 1: modified “a blood samples” to “blood samples”

POINT 2: Line 85 what was the final concentration of DNA added to each reaction?

Response 2: The DNA is attached to the FTA card, so we do not know its concentration, which instead is a function of the number of nucleated cells attached to the FTA card

POINT 3: Line 102 Add ‘by Sanger sequencing’ after ‘sequenced’ assuming this was the sequencing method used.

Response 3: Yes Sanger sequencing – wording amended.

POINT 4: Line 136 The analysis here uses the Texel Oar_v4 assembly, there is a newer more contiguous assembly for sheep Rambouillet v1 GCA_002742125.1, have the authors compared with this assembly? Presumably the difference if not related to the genome assembly could be a breed specific difference?

Response 4: No – we haven’t done that comparison, but we agree with the comment and have now run the analysis against the Rambouillet sequence.

POINT 5: Line 194 Could the authors clarify here in text that they are using the Oarv4 assembly. They could also make the point that the 63bp sequence might be resolved in the new assembly but this needs to be checked.

Response 5: See above.

POINT 6: Line 231 Reference to figure 3 could be removed.

Response 6: Deleted it as required by the reviewer.
